# Snoring and environmental exposure: results from the Swedish GA2LEN study

Daniel Silverforsen,[1] Jenny Theorell-Haglöw,[1] Mirjam Ljunggren,[1]
Roelinde Middelveld,[2] Juan Wang,[3] Karl Franklin,[4] Dan Norbäck,[3] Bo Lundbäck,[5]
Bertil Forsberg,[6] Eva Lindberg,[1] Christer Janson [1]

► Prepublication history and additional online supplemental material for this paper are available online. To view these files, please visit the journal online (http://dx.doi.org/10.1136/bmjopen-2020-044911).

¹Department of Medical Sciences, Respiratory, Allergy and Sleep Research, Uppsala Universitet, Uppsala, Sweden
²The Centre for Allergy Research and Institute of Environmental Medicine, Karolinska Institutet, Stockholm, Sweden
³Department of Medical Sciences, Environmental and Occupational Medicine, Uppsala Universitet, Uppsala, Sweden
⁴Department of Surgery, Umea Universitet, Umea, Sweden
⁵Krefting Research Centre, Goteborgs Universitet, Goteborg, Sweden
⁶Public Health and Clinical Medicine, Umea Universitet, Umea, Sweden

**Correspondence to**
Dr Christer Janson;
christer.janson@medsci.uu.se

## ABSTRACT:

**Objective** Habitual snoring is associated with fatigue, headaches and low work performance. This cross-sectional study aimed to investigate if snoring is affected by environmental factors such as home dampness and exposure to air pollution.

**Setting** General population sample from four Swedish cities.

**Participants** 25 848 participants from the Swedish part of the epidemiological Global Asthma and Allergy and European network of excellence study carried out in 2008. The participants completed a postal questionnaire on snoring and, indoor and outdoor environmental exposure as well as potential confounders including smoking, weight, height and educational level.

**Results** Of the participants, 4211 (16.3%) were habitual snorers. Habitual snorers reported water damage (8.3% vs 7.0% p<0.0001), floor dampness (4.6% vs 3.8% % p<0.0001) and visible mould (5.2% vs 3.8% p<0.0001) in their homes more often than non-snorers. Habitual snorers stated being annoyed by air pollution more often than non-snorers with habitual snorers reporting being irritated with the air in their residential area to a higher extent (sometimes 16.2% vs 13.9%, and daily 4.6% vs 3.1%) as well as annoyance from traffic fumes (somewhat 19% vs 18.5% and very 5% vs 3.6%) (p<0.0001). These results remained significant after adjustment for age, body mass index, smoking history and educational level.

**Conclusion** Snoring is more prevalent in subjects reporting home dampness and air pollution. These association should be confirmed in further research using objective measurements and a longitudinal approach.

## INTRODUCTION

Snoring is a sign of compromised upper airway and a common symptom in obstructive sleep apnoea (OSA).[1] Snoring is a health problem also in the absence of sleep apnoea, and studies in both adults and children show that it is the frequency of snoring rather than the Apnoea–Hypopnoea Index (AHI) that predicts symptoms and poorer behavioural and cognitive outcomes.[2 3]

Habitual snoring is reported to occur in between 16%–33% in men and 6%–19% in women.[4–8] Male sex, age, obesity, smoking and nasal congestion are all well-established

### Strengths and limitations of this study

► The sample size is large giving it considerate power and the possibility to simultaneously control for several confounding risk factors.
► The association between snoring and both indoor and outdoor exposure has only been studied to a limited extent previously.
► The study is cross-sectional and causality can therefore not been assessed.
► The assessments of exposure are self-reported and we know that there is only a moderate association between self-reported and measured air pollution.

risk factors for habitual snoring.[9–14] An association between home dampness and insomnia symptoms was reported in a population-based cross-sectional study,[15] but only a few studies have, to our knowledge investigated the possible association between habitual snoring and building dampness.[16]

Air pollutants such as particulate matter, ozone ($O_3$) and nitrogen dioxide all have adverse effects on the airways.[17] An association between exposure to air pollutants and snoring have been reported in children[18 19] and habitual snoring was associated with having a bedroom with traffic noise but not to high exposure to traffic pollution in a Nordic population study .[20] In another study, exposure to $O_3$ was associated with an increase in AHI.[21]

The aim of this study was to investigate if habitual snoring is affected by environmental factors. We hypothesised that both exposures to perceived building dampness and air pollution are associated with habitual snoring.

## METHOD

The Global Asthma and Allergy and European network of excellence (GA²LEN) was established 2004 as a consortium in the field of asthma and allergy.[22] The GA²LEN epidemiological study was carried out in 2008 across Europe, and the aim was to assess

the prevalence of allergic disease and its associated risk factors.[23] The present investigation was based on the Swedish part of the postal survey carried out in four centres; Stockholm, Gothenburg, Uppsala and Umeå. A random sample of 15 000 subjects from Gothenburg and 10 000 from each of the other cities was selected, and the participants were 16–75 years old.[24] The questionnaire included questions on asthma, chronic bronchitis, eczema, insomnia, OSA, physical activity and environmental and workplace exposure. Up to three reminders were mailed, and 26 577 subjects completed the questionnaire. In the present analysis, we included the 25 848 participants who had answered both the question regarding snoring and environmental exposure. Informed consent was obtained from each participant.

### Definitions

Snoring was assessed by the question, 'in the last few months, how many times have you snored loudly and disturbingly'. Habitual snorers were defined as having answered that they snored either 3–5 times per week or every or almost every night.

Smoking history was assessed by the questions 'Have you ever smoked one or more cigarettes per day for more than one year?' Those that answered yes to that question were then asked: 'Have you smoked at all during the last month?'.

Never smoking was defined as a negative response to the question, 'Have you ever smoked for at least one year?'.

Previous smoking was defined as a positive response to the question 'Have you ever smoked for at least one year?' and a negative response to the question 'Have you smoked at all during the last month?'.

Current smoking was defined as positive responses to the questions 'Have you ever smoked for at least one year?' and 'Have you smoked at all during the last month?'.

Air pollution was assessed by two questions 'Do you find the air irritating in your residential area: (1) daily/almost daily (daily), (2) sometimes/periodically (sometimes), (3) seldom/never (never) and 'How annoyed are you by fumes from traffic in your residential area.': (1) none/a little (none), (2) somewhat and (3) very much.

Dampness was defined as a positive response to at least one of the following questions: 'Have any of the following been noted in your home during the last 12 months:
1. Water/moisture damage on indoor walls, floors or ceiling? (Water damage)
2. Dented plastic mats, yellowed plastic mats or blackened parquet? (Floor dampness)
3. Visible moulds on walls, floors or ceiling?' (Visible moulds)

Educational level was assessed, and the subjects were divided into three groups depending on the highest academic level: primary school, high school, university.

Body mass index (BMI) was calculated from the subjects answer to their height and weight.

### Statistical analysis

The statistics where analysed using STATA V.15 (StataCorp). $\chi^2$ test and unpaired t-test were used in unadjusted analyses when comparing habitual snorers and non-snorers. Multiple logistic regression analysis was performed when studying the independent association between habitual snoring and environmental exposure. To decide which variables to adjust for in the multiple logistic regression models, a directed acyclic graph was used (http://www.dagitty.net) (online supplemental figure 1). The confounding factors used in the model were age, BMI, smoking history and educational level. Test for interaction was done for building dampness and air pollution indicators in the relationship with snoring. We also used a test for interaction to determine if there was a sex difference in the association between snoring and environmental variables. A p<0.05 was considered to be statistically significant.

### Patient and public involvement

The European Federation of Allergy and Airways Diseases Patients' Associations is a partner of the GA²LEN. Apart from this no patients or members of the public were involved in the design, conduct or dissemination plan for this study.

### RESULTS

Of the 25 848 participants 4211 (16.3%) were habitual snorers. Habitual snoring was more common in men and in older participants (table 1). Habitual snorers had a higher mean BMI and were less likely to have finished higher education, and were more likely to be ex-smokers and current smokers than non-snorers (table 1).

Habitual snorers reported water damage, floor dampness and visible mould more often than non-snorers. The habitual snorers also reported being exposed to air pollution more often than non-snorers (table 1).

Residential water damage, floor dampness and visible moulds were significantly associated with habitual snoring after adjusting for confounders (age, BMI, smoking history and educational level) (table 2). There was also an association between reported exposures to residential air pollution and being exposed to traffic related air pollution and habitual snoring, respectively, that remained statistically significant after adjustment for confounders (table 2).

There was no significant difference in the association between reported air pollution and habitual snoring in those that reported or did not report residential water damage. There was no significant difference between women and men in the associations between building dampness and air pollution with snoring (all $p_{interaction}>0.05$).

### DISCUSSION

This study's main result was that both self-reported home dampness and exposure to air pollution were significantly associated with habitual snoring. The results remained

**Table 1** Characteristics and environmental exposure in non-snorers and snorers (% and mean±SD)

| | Non-snorers (n=21 637) | Habitual snores (n=4211) | P value |
|---|---|---|---|
| Men | 42.3 | 62.6 | <0.0001 |
| Age | 42.1±16.0 | 50.0±13.9 | <0.0001 |
| Smoking history | | | <0.0001 |
| Never | 63.7 | 45.7 | |
| Ex | 23.6 | 34.2 | |
| Current | 12.7 | 20.1 | |
| BMI | 24.3±3.9 | 27.1±4.6 | <0.0001 |
| Educational level | | | <0.0001 |
| Primary school | 12.4 | 18.5 | |
| Secondary school | 28.5 | 35.8 | |
| Tertiary School | 59.1 | 45.8 | |
| Indoor dampness | | | |
| Water damage | 7.0 | 8.3 | 0.002 |
| Floor dampness | 3.8 | 4.6 | 0.02 |
| Visible moulds | 3.8 | 5.2 | <0.0001 |
| Air pollution in residential area | | | <0.0001 |
| Never | 83.0 | 79.2 | |
| Sometimes | 13.9 | 16.2 | |
| Daily | 3.1 | 4.6 | |
| Air pollution from traffic | | | <0.0001 |
| None | 77.9 | 76.0 | |
| Somewhat | 18.5 | 19.0 | |
| Very much | 3.6 | 5.0 | |

BMI, body mass index.

**Table 2** Association between environmental exposure and habitual snoring

| | OR (95% CI) | P value |
|---|---|---|
| Indoor dampness* | | |
| Water damage | 1.29 (1.13 to 1.48) | <0.0001 |
| Floor dampness | 1.21 (1.01 to 1.45) | 0.04 |
| Visible moulds | 1.51 (1.27 to 1.80) | <0.0001 |
| Air pollution in residential area† | | |
| Never | 1 | |
| Sometimes | 1.29 (1.13 to 1.48) | <0.0001 |
| Daily | 1.64 (1.36 to 1.98) | <0.0001 |
| Air pollution from traffic† | | <0.0001 |
| None | 1 | |
| Somewhat | 1.15 (1.04 to 1.26) | 0.004 |
| Very much | 1.51 (1.27 to 1.80) | <0.0001 |

*Adjusted for age, BMI, smoking history, educational level and residential air pollution.
†Adjusted for age, BMI, smoking history, educational level and residential water damage.
BMI, body mass index.

significant after having adjusted for age, BMI, smoking history and educational level.

The results that habitual snoring was associated with both airway pollution and home dampness are important as previous studies have found that habitual snores have an increased risk of work-related accidents,[25] daytime sleepiness,[26] hypertension,[27] diabetes[28] and increased mortality.[29] Some of these associations are probably related to obstructive sleep apnoea, but several studies have found that snoring, even in the absence of sleep apnoea, is associated with daytime sleepiness.[3 30] These results further implicate the importance of good air quality in home and outdoor environment for health.

Of the indoor dampness indicators, visible mould had the highest OR for habitual snoring, followed by water damage and floor dampness. As far as we know, a relationship between home dampness and snoring has only been in a few studies.[16] However, previous studies have shown associations between snoring and other respiratory conditions such as asthma and chronic rhinosinusitis.[31–33]

Zhang et al[34] showed an association between building dampness and non-respiratory symptoms such as eye and skin symptoms, headache and fatigue. An association between indoor dampness and insomnia symptoms has also been reported.[15] Headache and fatigue are symptoms that are associated with snoring.[10 35] It is, therefore, possible that the association between home dampness, fatigue and headache partly is related to a higher prevalence of habitual snoring in individuals living in damp homes.

In the present study, the level of annoyance was used as a proxy for the actual extent of pollution. The association between air pollution and snoring in our study is in accordance with previous research by Weinreich et al,[21] where there was an association between $O_3$ levels and obstructive sleep apnoea. In another study, children residing in the neighbourhoods with the poorest air quality had a significantly higher prevalence of habitual snoring than those who lived in the neighbourhoods with lower air pollution degrees.[18] Also, passive and active smoking has been associated with habitual snoring,[13] and air pollution and cigarette smoke share some chemical components and mechanic pathways.[36] However, our finding is not in line with the findings of Gislason et al,[20] were traffic noise but not high air pollution exposure was associated with snoring.

The prevalence of habitual snoring was 16% in our study sample. This is lower compared with a similar study on a Scandinavian population, which showed a prevalence of 25%.[20] As in many previous studies, habitual

snoring was associated with older age, male sex, higher BMI and smoking.[9]

The biological explanation for the association between snoring and environmental exposure in our study is not completely known. Home dampness and exposure to airway pollution has been associated with inflammatory markers in nasal lavage.[37 38] It is therefore possible that inflammation in the upper airways caused by home dampness and air pollution explains the association with snoring.

The study has a large sample size, thereby giving it considerate power and the possibility to simultaneously control for several confounding risk factors. However, there are weaknesses to be considered. The exposure assessments are self-reported, and we know that there is only a moderate association between self-reported and measured air pollution.[39] It is difficult to say how accurate the subjects' assessment of their snoring habits are, and it has been shown that there are some discrepancies between self-reported and spouse-reported snoring habits.[40] We also have no data on if the subjects live by themselves or not, a factor that may affect individuals knowledge of their snoring. The participation rate was 57%, and we know from previous analysis that younger persons and men are somewhat underrepresented in our population.[41] We do not, however, think that this has influenced the results in a major way.

In conclusion, home dampness and exposure to air pollution were significantly associated with habitual snoring also after adjusting for cofounders. These associations should be confirmed in further research using objective measurements and a longitudinal approach.

**Contributors** CJ is the corresponding author and DS and CJ drafted the paper and performed the statically analysis, CJ, RM, KF, BF and BL participated in designing the study and reviewed the manuscript; JT-H, ML, RM, JW, KF, DN. BL, BF and EL also reviewed the paper at several stages.

**Funding** This work has been supported by supported by the EU FP6 project GA2LEN (EU contract nr. FOOD-CT-2004-506378), the Centre for Allergy Research at the Karolinska Institutet, the Swedish Heart Lung Foundation (20090526) the Swedish Heart and Lung Association and the Swedish Asthma and Allergy Association.

**Competing interests** None declared.

**Patient consent for publication** Not required.

**Ethics approval** The study was approved by the Regional Ethical Review Board in Uppsala, Sweden approval (Dnr 2008/014).

**Provenance and peer review** Not commissioned; externally peer reviewed.

**Data availability statement** Data are available on reasonable request. Requests for data should be sent to the corresponding author.

**ORCID iD**
Christer Janson http://orcid.org/0000-0001-5093-6980

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
