## [Reviewer comments · BMJ Open]

ARTICLE DETAILS

TITLE (PROVISIONAL)	Snoring and environmental exposure: results from the Swedish GA2LEN study.
AUTHORS	Silverforsen, Daniel; Theorell-Haglöw, J; Ljunggren, Mirjam; Middelveld, Roelinde; Wang, Juan; Franklin, Karl; Norbäck, Dan; Lundbäck, Bo; Forsberg, Bertil; Lindberg, Eva; Janson, Christer

VERSION 1 – REVIEW

REVIEWER	Chamara Senaratna The University of Melbourne, Australia and The University of Sri Jayewardenepura, Sri Lanka
REVIEW RETURNED	17-Nov-2020

GENERAL COMMENTS	This study examined the association between environmental factors and snoring. It is an interesting area of research and help guide further research in this area. Some detailed comments are given below. Abstract This may have to change based on the suggested changes given below for analysis and results/discussion. Methods Definitions: I understand what is meant by current, previous and never smoking, but not sure what is meant by 'smoking' [line numbers 31-33]. Did the authors mean 'ever smoked'? The definition of never-smoked is not captured by the question asked. This question captures those who have never smoked as well as those who have smoked but not continuously for one year. It may be more meaningful to use a different term than 'never-smoked' to identify this group. It would be good to be consistent in the use of terminology – in the aims of the research the authors indicate that the association is measured with air pollution but in the methods, this is replaced with 'annoyance due to air pollution'. As this is the only way the air pollution can be measured in large population studies without high costs, I suggest keeping this as 'air pollution' in the methods section too, and maybe indicate that the level of annoyance was used as a proxy for the actual extent of pollution. Statistical analysis: It is unclear the authors' thinking behind choosing confounding variables to control for. I believe that associations that were checked were between environmental factors and snoring. It is easy to understand how factors such as sex, BMI, age etc could be associated with snoring given the current knowledge, but it is unclear how these factors also affect environmental factors. For a variable to be a confounder, it must be associated with both the exposure and the outcome. It may be
--

	good if the authors can illustrate these perceived associations in the directed acyclic graph [DAG] and include it in a supplement for clarity and to provide justifications for their choice of confounders. Results Tables 1 and 2 can possibly be combined in to one, with two sections within the table. I'm not sure if the Figure-1 is the best method to present the findings. These findings are important, and I prefer to see these in actual numbers in a table rather than in a figure like this. A table that shows the OR and CIs are clearer and probably easier to read than this figure. Besides, this figure in its current format doesn't help much to understand the interaction effects. Interaction effects given in the text can be shown in a table. The 4th paragraph under the Results section is unclear [line #s 33-35]. Suggest rewording for clarity. Do the authors have data on other respiratory conditions such as rhinosinusitis? If so, any mediation effects can be explored which could be interesting. Discussion In general, the discussion contains relevant facts but seems to hang loose without a clear focus. Suggest some re-writing to make it more focussed on the implications of the results and to be punchy. I realise that these implications are already included. However, these seem to have drowned in other more general explanations. It would also be good to include the effect of the high non-response rate. Do the authors have any information on the non-responders to compare with those who responded? It would be good to discuss implications of any selection bias. Page 8 line# 59: The explanation related to older population is unclear. The last sentence of the discussion [in conclusions] can be replaced with one that gives directions for further research. The current one sounds rather boring and non-specific.
--	---

REVIEWER	Prof Keith Morris Cardiff Metropolitan University, Cardiff, UK
REVIEW RETURNED	26-Dec-2020

GENERAL COMMENTS	The major strength of this study is that it is a relatively large study investigating proposed links between certain environmental factors and snoring. The authors undertake a detailed analysis of certain confounders associated with snoring and the analysis is appropriate, with Odds Ratios, their confidence intervals and significance reported. My main concern with the study is its almost total reliance on self-reporting of the main variables. There is a substantial interest in the links between snoring and several diseases and much reporting of it and the authors report significant Odds and their Confidence Intervals that do not cross unity. They are clear in their limitations however, I remain bemused how total reliance on self-reporting of snoring can be relied on, a substantial limitation by their own admission, or the measures of . Why do the authors not assess the reliability or validate the reporting of snoring, in subjects with partners who could be identified, or the assessment of the environmental factors by investigating a sub-group?
---

	Hence, I have no statistical issues of this study, but I am uncomfortable with the level of self-reporting, not in itself, but given the nature of the studies outcomes, I find these very limiting
--	---

VERSION 1 – AUTHOR RESPONSE

Reviewer: 1

Dr. Chamara Senaratna, University of Melbourne School of Population and Global Health Comments to the Author:

This study examined the association between environmental factors and snoring. It is an interesting area of research and help guide further research in this area. Some detailed comments are given below.

Abstract

This may have to change based on the suggested changes given below for analysis and results/discussion.

A: Changes have been made in the conclusion

Methods

Definitions: I understand what is meant by current, previous and never smoking, but not sure what is meant by ‘smoking’ [line numbers 31-33]. Did the authors meant ‘ever smoked’? The definition of never-smoked is not captured by the question asked. This question captures those who have never smoked as well as those who have smoked but not continuously for one year. It may be more meaningful to use a different term than ‘never-smoked’ to identify this group.

A: We’ve changed the term smoking to smoking history. We’ve also changed the order of the definitions of never, previous and current smokes. We agree that the definition of never-smokers will include some subjects that have smoked to a limited degree., However, this definition has been used in a large number of previous papers using the GA2LEN data and also studies with similar methodology such as ECRHS ad BOLD:

It would be good to be consistent in the use of terminology – in the aims of the research the authors indicate that the association is measured with air pollution but in the methods, this is replaced with ‘annoyance due to air pollution’. As this is the only way the air pollution can be measured in large population studies without high costs, I suggest keeping this as ‘air pollution’ in the methods section too, and maybe indicate that the level of annoyance was used as a proxy for the actual extent of pollution.

A: We agree and have changed the text accordingly (Methods, page 5, second last paragraph, Results page 7, third paragraph and Discussion page 8, fourth paragraph))

Statistical analysis: It is unclear the authors’ thinking behind choosing confounding variables to control for. I believe that associations that were checked were between environmental factors and snoring. It is easy to understand how factors such as sex, BMI, age etc could be associated with snoring given the current knowledge, but it is unclear how these factors also affect environmental factors. For a variable to be a confounder, it must be associated with both the exposure and the outcome. It may be good if the authors can illustrate these perceived associations in the directed acyclic graph [DAG] and include it in a supplement for clarity and to provide justifications for their choice of confounders.

A: A directed acyclic graph is now included as a supplement. It is true that smoking, age and BMI

doesn't have a direct effect on air pollution or dampness but they do have an effect on educational level which in its turn has an effect on the two exposure variables. We agree that sex is not a true confounder and have therefore excluded sex from the model.

Results

Tables 1 and 2 can possibly be combined in to one, with two sections within the table.

I'm not sure if the Figure-1 is the best method to present the findings. These findings are important, and I prefer to see these in actual numbers in a table rather than in a figure like this. A table that shows the OR and CIs are clearer and probably easier to read than this figure. Besides, this figure in its current format doesn't help much to understand the interaction effects. Interaction effects given in the text can be shown in a table.

A: Excellent suggestions. Table 1 and 2 have now been merged and figure 1 one has been turned into table 2.

The 4th paragraph under the Results section is unclear [line #s 33-35]. Suggest rewording for clarity.

A: We have now rewritten this paragraph in order to make it clearer.

Do the authors have data on other respiratory conditions such as rhinosinusitis? If so, any mediation effects can be explored which could be interesting.

A: That's an interesting idea. We have now done interaction analysis to see if the association between the environmental factors and snoring differed between those participants that did or did not have rhinitis but found no differences. We have not included this in the manuscript.

Discussion

In general, the discussion contains relevant facts but seems to hang loose without a clear focus. Suggest some re-writing to make it more focussed on the implications of the results and to be punchy. I realise that these implications are already included. However, these seem to have drowned in other more general explanations.

A: We have rewritten the discussion and think that the focus has improved.

It would also be good to include the effect of the high non-response rate. Do the authors have any information on the non-responders to compare with those who responded? It would be good to discuss implications of any selection bias.

A: We've added information on participation and possible influence of this in the discussion (page 9, paragraph 3).

Page 8 line# 59: The explanation related to older population is unclear.

A: Yes we agree and have deleted that part of the sentence

The last sentence of the discussion [in conclusions] can be replaced with one that gives directions for further research. The current one sounds rather boring and non-specific.

A: The conclusion has rewritten both in the abstract and the discussion section.

Reviewer: 2

Dr. Keith Morris, Cardiff Metropolitan University Comments to the Author:

The major strength of this study is that it is a relatively large study investigating proposed links between certain environmental factors and snoring. The authors undertake a detailed analysis of certain confounders associated with snoring and the analysis is appropriate, with Odds Ratios, their confidence intervals and significance reported.

A: Thank you

My main concern with the study is its almost total reliance on self-reporting of the main variables. There is a substantial interest in the links between snoring and several diseases and much reporting of it and the authors report significant Odds and their Confidence Intervals that do not cross unity. They are clear in their limitations however, I remain bemused how total reliance on self-reporting of snoring can be relied on, a substantial limitation by their own admission, or the measures of . Why do the authors not assess the reliability or validate the reporting of snoring, in subjects with partners who could be identified, or the assessment of the environmental factors by investigating a sub-group?

A: It is surprisingly difficult to record snoring and there are currently no standardised methods available. Nevertheless we have in previous studies that reported snoring is related to ad health effects such as accidents, daytime sleepiness, hypertension and diabetes. We have written a new paragraph in the discussion (page 8, paragraph 2).

Hence, I have no statistical issues of this study, but I am uncomfortable with the level of self-reporting, not in itself, but given the nature of the studies outcomes, I find these very limiting

A: Se comments above

Thank you for giving us a chance to improve our manuscript

VERSION 2 – REVIEW

REVIEWER	Senaratna, Chamara University of Melbourne School of Population and Global Health
REVIEW RETURNED	28-Mar-2021

GENERAL COMMENTS	The authors have addressed all of my concerns. The manuscript seems well written, and except for few grammar and spelling issues that are of editorial in nature, I have no further concerns. I wish to congratulate the authors for a work well done!
---

REVIEWER	Morris, Keith Cardiff Metropolitan University
REVIEW RETURNED	12-Mar-2021

GENERAL COMMENTS	I am now satisfied with the presentation, the analysis and its reporting of this paper
--